# Identification of Radiomic Signatures in Brain MRI Sequences T1 and T2 That Differentiate Tumor Regions of Midline Gliomas with H3.3K27M Mutation

**DOI:** 10.3390/diagnostics13162669

**Published:** 2023-08-14

**Authors:** Maria-Fatima Chilaca-Rosas, Manuel-Tadeo Contreras-Aguilar, Melissa Garcia-Lezama, David-Rafael Salazar-Calderon, Raul-Gabriel Vargas-Del-Angel, Sergio Moreno-Jimenez, Patricia Piña-Sanchez, Raul-Rogelio Trejo-Rosales, Felipe-Alfredo Delgado-Martinez, Ernesto Roldan-Valadez

**Affiliations:** 1Radiotherapy Department, Hospital de Oncología, Centro Medico Nacional Siglo XXI, Instituto Mexicano Del Seguro Social, Mexico City 06720, Mexico; maria.chilaca@imss.gob.mx (M.-F.C.-R.); david010198@hotmail.com (D.-R.S.-C.); 2Directorate of Research, Hospital General de Mexico Dr Eduardo Liceaga, Mexico City 06720, Mexico; melissa.garcialezama@gmail.com; 3Department of Internal Medicine, Hospital General de Mexico Dr Eduardo Liceaga, Mexico City 06720, Mexico; raul.vda1995@hotmail.com; 4Neurological Center, Neurosurgery Department of National Institute of Neurology and Neurosurgery, Mexico City 14269, Mexico; radioneurocirugia@gmail.com; 5Neurological Center, Neurosurgery Department of American British Cowdray Medical Center, Mexico City 01120, Mexico; 6Oncology Diagnostic, Unidad de Investigacion Medica en Enfermedades Oncologicas U.I.M.E.O, Hospital de Oncología, Centro Medico Nacional Siglo XXI, Instituto Mexicano Del Seguro Social, Mexico City 06720, Mexico; patricia_1307@yahoo.com.mx; 7Medical Oncology, Hospital de Oncología, Centro Medico Nacional Siglo XXI, Instituto Mexicano Del Seguro Social, Mexico City 06720, Mexico; rexiboro@gmail.com; 8Magnetic Resonance Service, Hospital de Especialidades, Centro Medico Nacional Siglo XXI, Instituto Mexicano del Seguro Social, Mexico City 06720, Mexico; felipolico@gmail.com; 9Department of Radiology, I.M. Sechenov First Moscow State Medical University (Sechenov University), 119992 Moscow, Russia

**Keywords:** midline gliomas, brain tumor, radiomics, MRI

## Abstract

Background: Radiomics refers to the acquisition of traces of quantitative features that are usually non-perceptible to human vision and are obtained from different imaging techniques and subsequently transformed into high-dimensional data. Diffuse midline gliomas (DMG) represent approximately 20% of pediatric CNS tumors, with a median survival of less than one year after diagnosis. We aimed to identify which radiomics can discriminate DMG tumor regions (viable tumor and peritumoral edema) from equivalent midline normal tissue (EMNT) in patients with the positive H3.F3K27M mutation, which is associated with a worse prognosis. Patients and methods: This was a retrospective study. From a database of 126 DMG patients (children, adolescents, and young adults), only 12 had H3.3K27M mutation and available brain magnetic resonance DICOM file. The MRI T1 post-gadolinium and T2 sequences were uploaded to LIFEx software to post-process and extract radiomic features. Statistical analysis included normal distribution tests and the Mann–Whitney U test performed using IBM SPSS^®^ (Version 27.0.0.1, International Business Machines Corp., Armonk, NY, USA), considering a significant statistical *p*-value ≤ 0.05. Results: EMNT vs. Tumor: From the T1 sequence 10 radiomics were identified, and 14 radiomics from the T2 sequence, but only one radiomic identified viable tumors in both sequences (*p* < 0.05) (DISCRETIZED_Q1). Peritumoral edema vs. EMNT: From the T1 sequence, five radiomics were identified, and four radiomics from the T2 sequence. However, four radiomics could discriminate peritumoral edema in both sequences (*p* < 0.05) (CONVENTIONAL_Kurtosis, CONVENTIONAL_ExcessKurtosis, DISCRETIZED_Kurtosis, and DISCRETIZED_ExcessKurtosis). There were no radiomics useful for distinguishing tumor tissue from peritumoral edema in both sequences. Conclusions: Less than 5% of the radiomic characteristics identified tumor regions of medical–clinical interest in T1 and T2 sequences of conventional magnetic resonance imaging. The first-order and second-order radiomic features suggest support to investigators and clinicians for careful evaluation for diagnosis, patient classification, and multimodality cancer treatment planning.

## 1. Introduction

Gliomas represent the most significant cause of cancer-related deaths in the population under 19 years [1], while brain tumors remain a substantial cause of morbidity in the population of adolescents and young adults (AYAs) defined by the National Cancer Institute as patients aged 15–39; corresponding to approximately 30% of CNS cancers [2]. Diffuse midline gliomas (DMG) represent about 20% of all pediatric tumors of the central nervous system (CNS). They are considered to have a devastating prognosis, with a median survival of less than one year after diagnosis [3,4].

Radiomics is an emerging field within imaging; it refers to acquiring quantitative features from medical images that are usually not perceptible to human vision. Subsequently, this information is transformed into high-dimensional data, which can eventually be extracted to find its relevance by correlating it with tumor histological characteristics, underlying genetic mutations, and malignancy, along with grade, progression, therapeutic effect, or even overall survival (OS) [5,6,7].

Radiomics studies in the specific context of midline gliomas are still in their beginning, with only three publications in PubMed at the time of writing this article (March 2023), including the terms “radiomics” AND “midline gliomas” in its title, the first from 2020 [8,9,10]. Most of these articles describe automatic machine techniques; however, a simple analysis of one-by-one radiomics about its usefulness in discriminating tumor regions is still missing in the literature.

In recent years, the role of different mutations has been studied with significant findings, such as the presence of H3.3K27M in most DMG, which is known to be associated with a worse prognosis [1,4,9]. Mackay et al. conducted a study with a cohort of >1000 patients in which the H3 mutant groups were compared, finding that the worst OS was associated with the H3.3K27M mutation, with a median of 11 months (whereas 18 months was associated with H3.3G34R/V and for H3.1/3.2K27M OS at H3.1/3.2K27M of 15 months). In addition, the H3.3K27M mutation had a 2-year OS of 4.7%. That is 5.8 times worse than the 27.3% 2-year OS of H3.3G34R/V [1]. Despite advances, no curative therapy has been found for DMG [11]. These antecedents led our interest to identify novel quantitative imaging biomarkers represented by radiomics that can complement the assessment of this pathology, allowing a deeper information acquisition than medical images alone. This is particularly helpful in cases where a biopsy cannot be performed, which is common in this disease. 

Our group recently published an article on the diagnostic performance of significant MRI-derived radiomics. However, we did not describe useful radiomics among individual tumor regions and statistical techniques and data visualization methods used to identify valid radiomic signatures [12].

Considering the information mentioned above, in this study, our objective was to identify radiomic signatures valid for discriminating tumor regions in DMG brain MRI sequences post-gadolinium T1 and T2, precisely characterizing patients with the H3.F3K27M mutation. Well respected software for radiomics calculation (LIFEx V.7.1.0) was used for this analysis, and our findings were described with comprehensive data visualization methods. We hope this study can contribute to reaching a consensus on the procedures for processing radiomics applicable to other CNS tumors.

## 2. Patients and Methods

### 2.1. Subject and Study Design

This study was an observational retrospective study approved by the National Committee for Scientific Research of Oncology and Pediatric Hospital of the National Medical Centre SXXI IMSS, Mexico City, with the number F-CNIC-2020-321. All the information reported is under national and international standards for the management of clinical files, the official Mexican norm NOM-012-SSA3-2012 [13], which establishes the criteria for the execution of scientific projects for human health, and with the Declaration of Helsinki of 1975, as revised in 2013 [14]. This study included patients with midline gliomas identified in pediatrics and AYA in the oncology and pediatric hospital database of the National Medical Centre S.XXI IMSS, Mexico City, from 2016 to 2021.

The inclusion criteria included a histopathological diagnosis of midline glioma with determination of the H3.3K27M mutation (Figure 1) according to WHO 2007 and 2016 (we did not use the WHO CNS 2021 criteria, as this study included patients from the 2016–2021 period before the WHO CNS5 publication) [15,16]. The second criterion was the location of the tumors in the midline region based on magnetic resonance imaging (MRI). The exclusion criteria were the concurrence of DMG with other malignancies in 5 years and the presence of another lethal comorbidity.

The description of patient selection criteria and histopathological diagnosis was previously described by our group in 2023 [12]. A flow diagram summarizes this process (Figure 2).

### 2.2. Imaging Evaluation of Midline Gliomas and Software for Calculation of Radiomics

Magnetic resonance examinations were performed on a 1.5 T magnetom (Siemens Healthcare, Erlangen, Germany) and a 3 T magnetom (Philips Health, Hong Kong, China). Tumor regions included viable tumors, peritumoral edema, and equivalent midline normal tissue (EMNT). The description of MRI evaluation was previously described in our publication in 2023 [12]. The images were not modified during preprocessing; however, they went through a selection process which included the mentioned sequences and slices where the region of interest (ROI) was visible. The images were then uploaded directly to the software LIFEx v.7.1.0 for post-processing [17].

We included 82 radiomic features in our calculation. In each of the selected brain magnetic resonance images, three regions were chosen to extract the radiomics: viable tumor (represented for enhancing tumor regions in T1 post-gadolinium sequence), peritumoral edema (visualized on T2), and equivalent midline normal tissue (EMNT) observed in both sequences. Two expert neuro-oncologists drew ROIs manually across all slices of the selected sequences individually (T1 post-gadolinium and T2), where the three tumor regions were present. Figure 3 shows an example of the MRI appearance of DMG.

### 2.3. Statistical Analysis

Considering that the main objective of this study was to identify which of the 82 radiomics helped differentiate tumor tissue from EMNT and peritumoral edema from EMNT in the two most common brain magnetic resonance sequences used to diagnose DMG (T1 post-gadolinium and T2), our assessment consisted of a three-step analysis.

In step 1, independent comparisons were performed between each radiomic and their corresponding tumor regions. As the normality test demonstrated a non-normal distribution for all radiomics using Kolmogorov–Smirnov and Shapiro–Wilk tests, subsequently, a non-parametric test (Mann–Whitney U) was used to compare the differences between the values of EMNT vs. tumor region and EMNT vs. peritumoral region. Significant statistical differences were demonstrated by a *p*-value < 0.05. Only radiomics that showed statistical differences between normal and abnormal tissue (viable tumor and peritumoral edema) were selected for each sequence.

In step two of our analyses, we selected only those radiomics that coincidentally could discriminate normal vs. abnormal tissue in T1 post-gadolinium and T2 sequences. Radiomics previously identified in T1 post-gadolinium and T2 were compared again using Venn diagrams [18]. We used closed circles drawn on a plane to illustrate with intuitive visualizations the relationships between two sets of selected radiomics in T1 post-gadolinium and T2, with a particular interest in its intersection. An additional analysis was performed to compare radiomics from viable tumor tissue vs. peritumoral edema in both sequences, using the Mann–Whitney U test to identify significant differences between these regions.

In step three, we included a quantitative assessment using the radiomic values of non-parametric statistics combined with a graphical component as part of a data visualization technique. We used a modification of the method known as analysis of means (ANOM), previously described in the quantitative evaluation of MRI biomarkers with apparent diffusion coefficient values in stroke [19], as a statistical approach to compare multiple means simultaneously [20]. As we had the quartiles (Q1, median, Q3) and interquartile ranges (IQR) of selected radiomics, we used a graphical representation of the median of medians analyses to represent the values of chosen radiomics.

### 2.4. Software

The database was created using Microsoft Excel^®^ v.16.67 (Microsoft Corporation, Redmon, WA, USA), and data visualization of Venn diagrams were done using Microsoft Power Point^®^ v.16.67 (Microsoft Corporation, Redmon, WA, USA). Statistical analyses were performed using IBM SPSS^®^ (Version 27.0.0.1, International Business Machines Corp.; Armonk, NY, USA). 

## 3. Results

### 3.1. Demographics and Clinical Features

Of the 12 patients included, 9 were pediatric (75%) and 3 were AYA (25%). The relationship between male and female patients (M:F) was 2:1. Tumors were located more frequently in the pons (50%). The rest of the tumors were found equally in the thalamus (25%) and midbrain (25%). These and other demographic characteristics can be found in Table 1.

### 3.2. Selected Radiomics Measurements in Post-Gadolinium T1 and T2 Sequences

From all selected regions (viable tumor, peritumoral edema, and EMNT), it was possible to extract 82 valuable radiomic features for each tumor region. These measurements represented 492 radiomic characteristics per patient (246 for each sequence, T1 after gadolinium and T2), a total of 5904 radiomic values.

For the T1 sequence, only ten radiomics showed a statistically significant difference (*p*-value < 0.05) between the viable tumor and the EMNT regions. Simultaneously, radiomics from peritumoral edema and EMNT were also compared in T1 post-gadolinium, obtaining five radiomics with statistically significant differences (*p*-value < 0.05). Table 2 shows a list of radiomics in the post-gadolinium sequence T1 that were able to discriminate pathological tissue from EMNT. Figure 4 shows a graphical representation of a median of median analyses, showing the values of selected radiomics in T1 post-gadolinium.

For the T2 sequence, 14 radiomics between viable tumor and EMNT regions showed a statistically significant difference (*p*-value < 0.05). Simultaneously, radiomics from peritumoral edema and EMNT were also compared, obtaining four radiomics with statistically significant differences (*p*-value < 0.05). Table 3 shows a list of the radiomics selected from T2 that were able to discriminate pathological tissue from EMNT. Figure 4 shows a graphical representation of a median of median analyses showing the values of chosen radiomics in T2. Figure 5 shows the radiomics data that discriminate between tumor regions using Venn diagrams. 

### 3.3. Identification of Radiomics Useful for Discriminating DMG in T1 and T2 Sequences

First, we compared the radiomics obtained analyzing EMNT vs. Tumor; of the 10 radiomics identified in the T1 post-gadolinium sequence and the 14 radiomics identified in the T2 sequence, only 1 radiomic was selected because it coincidentally appeared in both sequences (DISCRETIZED_Q1).

Similarly, the radiomics obtained in T1 post-gadolinium and T2 sequences from peritumoral edema and EMNT were compared; of the five radiomics identified in the T1 post-gadolinium sequence and the four radiomics identified in the T2 sequence, four radiomics coincided in both sequences and were then selected (CONVENTIONAL_Kurtosis, CONVENTIONAL_ExcessKurtosis, DISCRETIZED_Kurtosis, and DISCRETIZED_ExcessKurtosis). Figure 6 shows the names of radiomics useful for discriminating simultaneously (in T1 post-gadolinium and T2) tumor regions.

### 3.4. Comparison of Viable Tumor versus Peritumoral Edema

An additional evaluation was performed to compare both pathological tissues. In an initial step, radiomics from each region (viable tumor and peritumoral edema) from the post-gadolinium sequence T1 were compared, obtaining only two radiomics (CONVENTIONAL_Skewness and DISCRETIZED_Skewness) with statistically significant differences (*p*-value < 0.05). Simultaneously, radiomics from the T2 sequence in both pathological tissues were compared obtaining only one radiomic (GLRLM_LRLGE) with a statistically significant difference (*p*-value < 0.05). No radiomics were found to coincidentally discriminate a viable tumor from peritumoral edema in both sequences (T1 post-gadolinium and T2). Upon a reasonable request, the corresponding author can provide the complete list of the 492 radiomic values per patient.

## 4. Discussion

In this study, we identified radiomics that could discriminate different regions of DMG tumors (EMNT, viable tumor, and peritumoral edema) in MRI-derived T1 and T2 sequences. We could summarize our findings into six components. First, using non-parametric tests, we proved that in the post-gadolinium T1 sequence, only ten radiomics helped identify viable tumors from EMNT, and five radiomics distinguished peritumoral edema from EMNT. Second, with the same method for the T2 sequence, we found 14 radiomics that could distinguish tumor tissue from EMNT and four radiomics applicable to peritumoral edema from EMNT. Third, only 1 out of 82 radiomics (1.22%) could simultaneously characterize viable EMNT tumors in T1 post-gadolinium and T2 sequences.

Furthermore, only 4 out of 82 radiomics (4.88%) differentiated peritumoral edema from EMNT using the post-gadolinium T1 and T2 MRI sequences. Fourth, in evaluating tumor tissue versus peritumoral edema, no radiometric was found helpful in distinguishing these regions in both sequences. Fifth, we supplemented our statistical analyses with Venn diagram data visualization and median analysis (data visualization techniques have recently been used to assess quantitative MRI biomarkers [21]). Sixth, we obtained critical evidence that non-parametric methods retrieved a different list of radiomics; compared to the diagnostic performance technique against overall survival (OS) and progression-free survival (PFS) in DMG, our group recently published the latter method using the same DMG database [12]. Our group previously described OS and PFS definitions [22,23]. Obtaining radiomic features that can discriminate between healthy tissue and tumor components could be very useful in multimodal treatment planning in clinical practice. This includes the evaluation of these significant radiomics in patients that cannot be biopsied, supporting the assessment and decision tree. In addition, it provided further information about the extension of tumor tissue that may not be visible to the human eye.

To our knowledge, three previous publications on radiomics and DMG [8,9,10] focused on machine learning and the MRI features of the tumors. However, the supportive approach to diagnosing specific tumor regions was not appreciated. Two publications, one from 2021 [24], focused on prediction, and another from 2022 used an MRI-based radiomics model to predict the mutant status of H3.3K27M [25]; the authors extracted radiomic features from 8 sequences and only 18 sets of components were analyzed. 

Our approach was straightforward and designed transparently with the intention that it could be repeated by researchers worldwide. We intentionally avoided using automatic machine learning techniques, as we consider that there is still a black box in its methods that limits its use to selected hospitals or research institutions. We agree with other authors that using radiomics will increase the value of imaging in the differential diagnosis of glioma and reveals that its implementation in clinical practice is still pendent. A recent systematic review of the current status and quality of radiomics for glioma differential diagnosis in 2022 showed that the radiomic quality score (RQS) of 42 studies was only 24.21%, which meant that current radiomic studies for glioma differential diagnosis still lack the quality required to allow its introduction into clinical practice [26,27]. We identified several research trends based on radiomics and gliomas (not only DMG), including construction using multiparametric magnetic resonance radiomics (several MRI sequences combined with genotype status and clinical features), [25,28,29]; PET-extracted radiomics [30,31,32,33]; radiomics-based machine learning [34,35,36]; predictive models of recurrence [37,38]; survival and classification in gliomas [39,40,41]; and differential diagnosis [42,43].

## 5. Limitations

Some limitations of this study need to be addressed. We used a small sample size of 12 patients and evaluated only the T1 post-gadolinium and T2 sequences available in the brain MRI of all patients. We did not use machine learning or convoluted neural networks for our assessment as they were outside the scope of our study. Evaluation of gliomas using advanced magnetic resonance biomarkers has been one of the leading research lines of our group in the last decade [22,44,45,46]. The ability of radiomics to differentiate DMG from other brain tumors and its comparison with other imaging methods, such as CT and PET, was also beyond the scope of this study. Although the software employed in this manuscript is free and, therefore, could be easily replicable in more institutions, it is essential to mention that there needs to be some training for the ROI selection process. Experts in this field are neuro-oncologists and neuroradiologists, although other medical professionals with adequate training in neuroanatomy could also perform this analysis. The LIFEx software is increasingly used in medical research; we found 16 glioma-related publications that performed radiomic analyses using this software [17].

The small sample size mentioned results from being an orphan entity and the lack of tumor banks for this pathology in our country for further molecular determinations. However, research in this field will not cease to increase, and this research will be part of essential and complex research that will follow.

## 6. Future Directions

The clinical applications of radiology in brain tumors are currently considered part of non-invasive detection techniques and have been grouped into three categories: detection, characterization, and predictive monitoring [47]. 

Three of the immediate challenges we identified in adopting radiomics are: first, it is essential to understand how selected radiomics extracted from brain MRIs correlate with other advanced quantitative MRI biomarkers in gliomas such as diffusion tensor imaging (DTI), apparent diffusion coefficient, and MRI spectroscopy. This knowledge will help us understand the correlations between radiomics and cancer outcomes; recent studies have revealed associations between fractional anisotropy and OS [44] and correlations between DTI and spectroscopy [23].

Second, imaging data are currently accepted as one of the five basic types of big data in cancer research [48]. Clinicians and researchers should know that data themselves are useless; the algorithms encoding causal reasoning and domain (clinical and biological) [49] of radiomic characteristics are based on the observed relationship between data-driven radiomics and visual image content [50].

Third, a global consensus on radiomics in clinical practice will bring together two critical concepts in modern radiotherapy: adaptive radiotherapy and biological targeting, such as dose paint; treatment plans might be updated regularly to accommodate the observed treatment response [51]. Radiomics techniques will be added to the advanced MRI techniques currently accepted for precision radiotherapy in glioblastoma (contrast MRI, arterial spin labelling, quantitative blood oxygen level-dependent MRI (qBOLD), M.R. spectroscopy, diffusion-weighted imaging, PET, and DTI) [52]. It would also be desirable to have a global consensus on survival modelling in which researchers could integrate radiomic values from tumor regions during consecutive brain MRIs that the patient with gliomas undergoes as part of their follow-up; previous predictive models of survival in gliomas have been published using quantitative biomarkers of MRI [53]. 

We identified two dominant research trends using radiomics applied to MRI in cancer: predictive models using multiregional radiomics [39] and multisequence MRI-based radiomic models [54]. We hope that convergence in these methods will be available in the short-term and there will be global consensus on their use.

## 7. Conclusions

Adopting radiomics in the characterization of DMG tumor habitat provided new knowledge about these tumors; however, this information also comes with unique challenges necessary for its applicability in neuro-oncology, worldwide adoption, and further understanding of the biological meaning of the radiomic features. The fact that less than 5% of 82 radiomics helped identify tumor regions in conventional MRI sequences is relevant and could aid researchers before feeding radiomics to automatic machine learning algorithms for massive diagnosis and classification of patients. As processing extensive image data requires computational power and consumes technological resources, a legal framework of regulations will be necessary to allow insurance companies to help defray the cost of these supplementary image evaluations in the short-term. 

It is essential to acknowledge that tumors such as DMG are very aggressive and progress rapidly, often not allowing a biopsy to be performed. In this scenario, a novel technique enabling information acquisition only with non-invasive procedures such as imaging opens the door to opportunity. It could provide a more reliable presumptive diagnosis in those cases where the definitive (histopathological) diagnosis cannot be determined. Since it does not require advanced imaging techniques nor highly advanced procedures, this analysis could also be performed before reaching tertiary care, permitting an opportune diagnosis and, therefore, allowing earlier referral and management.

The present study identified the tumor lesion from healthy tissues in T1 and T2 sequences with first-order radiomic characteristics. Specifically, in T2, it was found to discriminate tumor tissue and edema from equivalent healthy tissue with first- and second-order radiomic characteristics such as GLRLM, suggesting support for identifying and delimitating tumor burden and for decision trees or algorithms, such as machine learning platforms for diagnostic purposes and multimodal management such as surgery and radiotherapy.

## Figures and Tables

**Figure 1 diagnostics-13-02669-f001:**
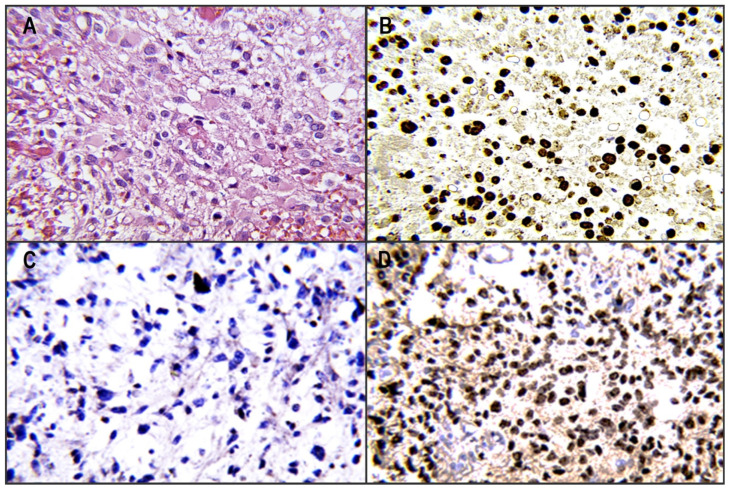
(**A**) Tumor tissue of a diffuse high-grade glioma with histological characteristics with hematoxylin-eosin staining (40×). (**B**) Tumor tissue of a pediatric patient with diffuse nuclear immunostaining with recognition of the mutated H3F3A K27M protein. (**C**) Tumor tissue negative for immunostaining with monoclonal antibody to express the mutated H3F3A K27M protein. (**D**) Positive immunostaining for the expression of the mutated protein H3F3A K27M in the tumor tissue of a young adult patient.

**Figure 2 diagnostics-13-02669-f002:**
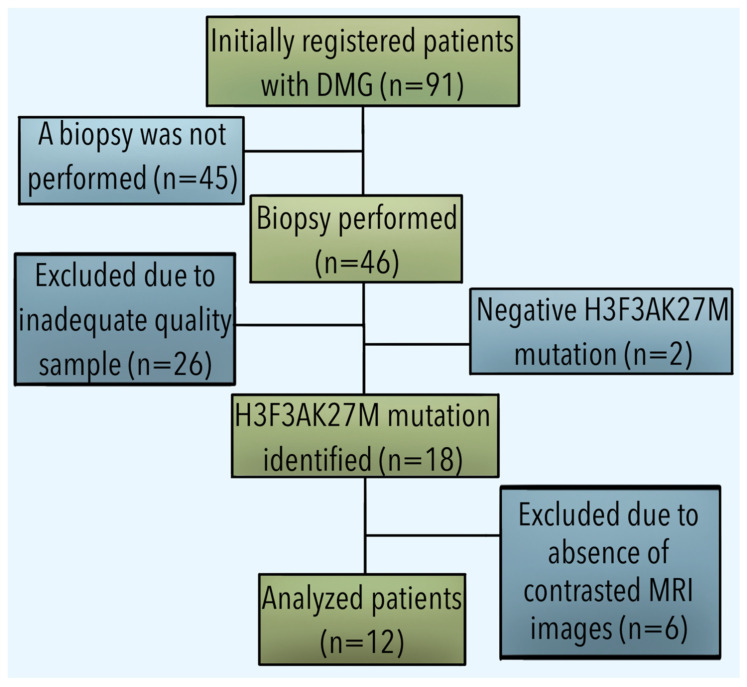
Flow diagram of the patient selection process.

**Figure 3 diagnostics-13-02669-f003:**
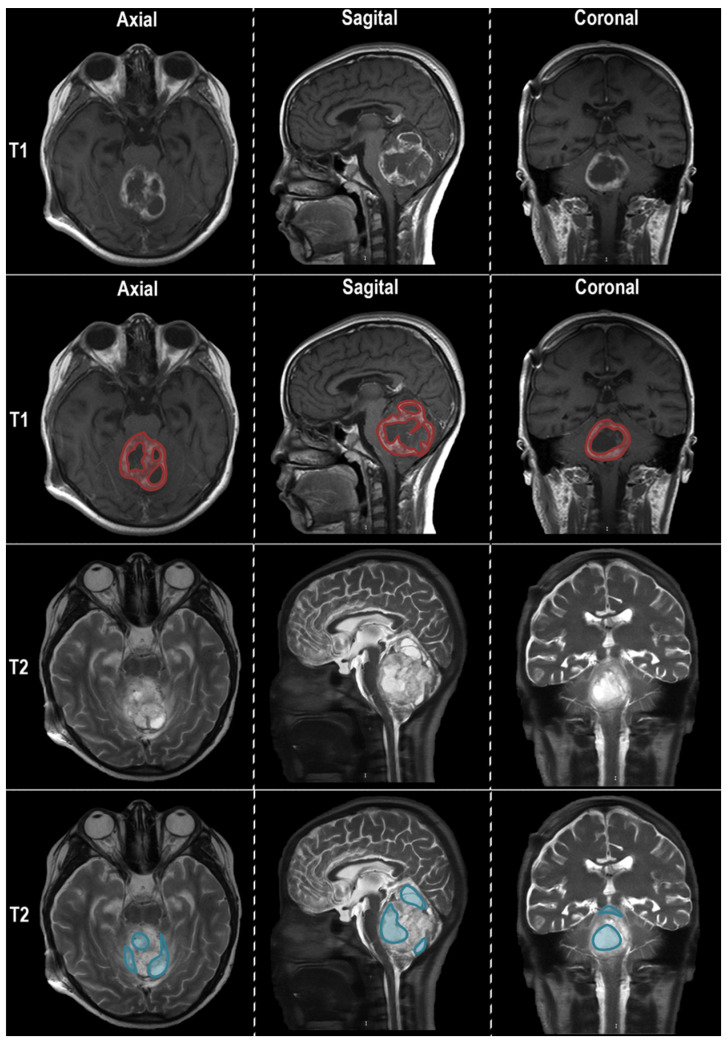
The MRI appearance of DMG on T1- and T2-weighted images is shown in three orthogonal planes (axial, sagittal, and coronal). The upper row depicts the T1 sequence tumor appearance. The second row represents the selection of the region of interest (ROI) for tumoral tissue (red) in the T1 sequence. T2 images provide better visualization of peritumoral edema (third row). The bottom row depicts the ROI for peritumoral edema (blue).

**Figure 4 diagnostics-13-02669-f004:**
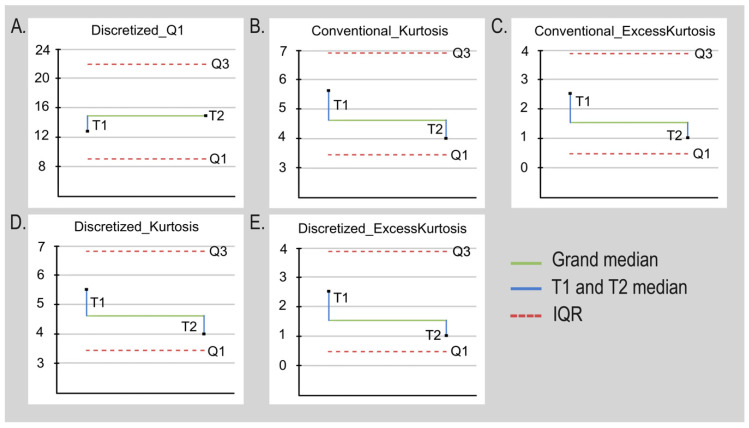
Graphical representation of a median analysis of medians of significant radiomics.

**Figure 5 diagnostics-13-02669-f005:**
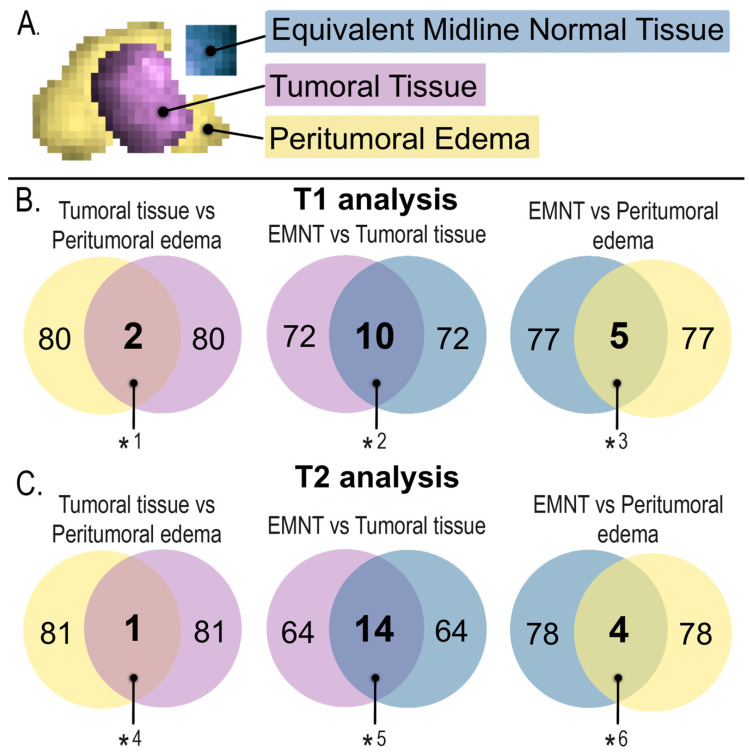
Presentation of 17 radiomics of statistical significance to discriminate between tumor regions in T1 and 19 radiomics in T2. Both sequences maintained a tendency for heterogeneity. (**A**) Graphical representation of the selected tumor regions: Equivalent midline normal tissue (blue), tumor tissue (purple), and peritumoral edema (yellow). (**B**) T1 sequence analysis, the different regions of tumors (tumoral tissue, peritumoral edema, and EMNT) were compared, selecting only radiomics that showed a statistically significant difference between regions *1: CONVENTIONAL_Skewness, DISCRETIZED_Skewness. *2: CONVENTIONAL_std, CONVENTIONAL_Skewness, CONVENTIONAL_Kurtosis, CONVENTIONAL_ExcessKurtosis, CONVENTIONAL_RIM_stdev, DISCRETIZED_Q1, DISCRETIZED_Skewness, DISCRETIZED_Kurtosis, DISCRETIZED_ExcessKurtosis, PARAMS_BinSize. *3: CONVENTIONAL_Kurtosis, CONVENTIONAL_ExcessKurtosis, DISCRETIZED_Kurtosis, DISCRETIZED_ExcessKurtosis, DISCRETIZED_HISTO_Energy[=Uniformity]. (**C**) The exact process was performed in the T2 sequence. *4: GLRLM_LRLGE. *5: DISCRETIZED_mean, DISCRETIZED_Q1, DISCRETIZED_Q2, DISCRETIZED_Q3, GLRLM_LGRE, GLRLM_HGRE, GLRLM_SRLGE, GLRLM_SRHGE, GLRLM_LRLGE, GLRLM_LRHGE, GLZLM_LGZE, GLZLM_HGZE, GLZLM_SZLGE, GLZLM_SZHGE. *6: CONVENTIONAL_Kurtosis, CONVENTIONAL_ExcessKurtosis, DISCRETIZED_Kurtosis, DISCRETIZED_ExcessKurtosis. Abbreviations: EMNT, equivalent midline normal tissue.

**Figure 6 diagnostics-13-02669-f006:**
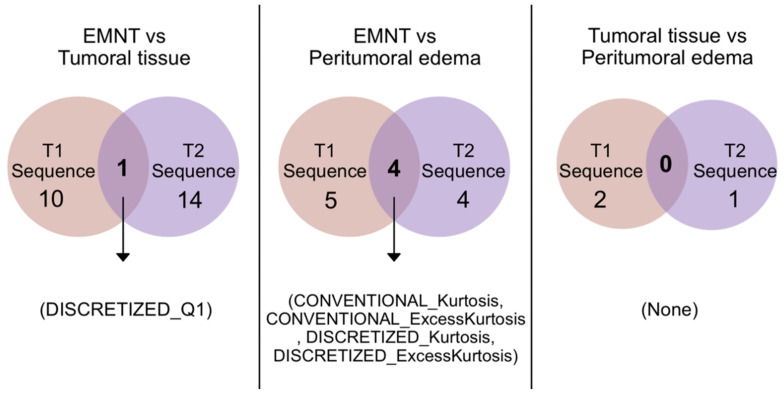
The radiomics previously identified with a statistically significant difference were then compared in the T1 (orange circle) and T2 (purple circle) sequences using a Venn diagram, obtaining the radiomics useful to distinguish the different regions of the tumor.

**Table 1 diagnostics-13-02669-t001:** Clinical characteristics of patients for radiomic analysis.

Clinical Characteristics	*n* = 12 (100%)
Age	Median ten years (range 19–29) years
Sex (M:F)	2:1
Paediatric patients	9 (75)
Young Adults Patients	3 (25)
Anatomical location	
Thalamus	3 (25)
Midbrain	3 (25)
Pons	6 (50)
Volume	Median 18.63 mL (range 4.82–140.15)
Volume voxel	Median 7368.5 voxels (range 1545–28,582)
KNF	
≥80	3 (25)
≤70	9 (75)
Surgical intervention	
Biopsy	10 (83.4)
STR	1 (8.3)
GTR	1 (8.3)
Radiation therapy dose	Median: 55 Gy (range 54–55 Gy)
Radiation therapy scheme	
Conventional	10 (83.3)
Hypofractionated	2 (16.7)
Chemotherapy	
Platinum-based with temozolomide	9 (75)
None.	3 (25)

Abbreviations: *n* = number of subjects studied, F = female, M = male, KNF: Karnofsky Functional Status Scale, Gy = Grey, mL = milliliters, STR = subtotal tumoral resection, GTR = gross tumoral resection.

**Table 2 diagnostics-13-02669-t002:** Radiomic values extracted from the T1 post-gadolinium sequence corresponding to the tumor regions that showed significant differences *.

	*n*	Percentile			IQR	*p*-Value
		Median	75	25		
** *Radiomics that showed a significant difference between EMNT vs. tumoral tissue* **						
CONVENTIONAL_std	24	64.756	123.139	40.629	82.51	0.003
2. CONVENTIONAL_Skewness	24	0.115	0.592	−1.122	1.714	0.027
3. CONVENTIONAL_Kurtosis	24	5.197	9.514	3.401	6.113	0.043
4. CONVENTIONAL_ExcessKurtosis	24	2.197	6.514	0.401	6.113	0.043
5. CONVENTIONAL_RIM_stdev	24	50.362	101.225	27.328	73.897	0.005
6. DISCRETIZED_Q1	24	13.000	18.750	9.250	9.5	0.047
7. DISCRETIZED_Skewness	24	0.120	0.592	−1.128	1.72	0.027
8. DISCRETIZED_Kurtosis	24	5.166	9.152	3.396	5.756	0.043
9. DISCRETIZED_ExcessKurtosis	24	2.166	6.152	0.396	5.756	0.043
10. PARAMS_BinSize	24	20.000	36.711	13.528	23.183	0.020
** *Radiomics that showed a significant difference between EMNT vs. peritumoral edema* **						
CONVENTIONAL_Kurtosis	24	5.197	9.514	3.401	6.113	0.017
2. CONVENTIONAL_ExcessKurtosis	24	2.197	6.514	0.401	6.113	0.017
3. DISCRETIZED_Kurtosis	24	5.166	9.152	3.396	5.756	0.017
4. DISCRETIZED_ExcessKurtosis	24	2.166	6.152	0.396	5.756	0.017
5. DISCRETIZED_HISTO_Energy[=Uniformity]	24	0.099	0.148	0.062	0.086	0.050
** *Radiomics that showed a significant difference between tumor tissue versus peritumoral edema* **						
CONVENTIONAL_Skewness	24	0.115	0.592	−1.122	1.714	0.028
2. DISCRETIZED_ Skewness	24	0.120	0.592	−1.128	1.72	0.028

IQR: Interquartile range; * Bivariate assessment used the Mann–Whitney U test.

**Table 3 diagnostics-13-02669-t003:** Radiomic values extracted from the T2 sequence corresponding to tumor regions showed significant differences *.

	*n*	Percentile			IQR	*p*-Value
		Median	75	25		
**Radiomics that showed a significant difference between EMNT vs. tumoral tissue**						
DISCRETIZED_mean	28	19.929	24.208	12.756	11.452	0.037
2. DISCRETIZED_Q1	28	16.000	21.500	11.000	10.5	0.037
3. DISCRETIZED_Q2	28	20.000	23.500	12.250	11.25	0.044
4. DISCRETIZED_Q3	28	23.500	27.500	15.000	12.5	0.044
5. GLRLM_LGRE	24	0.004	0.030	0.002	0.028	0.005
6. GLRLM_HGRE	24	435.251	672.443	135.606	536.837	0.004
7. GLRLM_SRLGE	24	0.004	0.027	0.002	0.025	0.009
8. GLRLM_SRHGE	24	405.484	604.182	127.266	476.916	0.005
9. GLRLM_LRLGE	24	0.007	0.050	0.004	0.046	0.007
10. GLRLM_LRHGE	24	592.797	1021.915	181.131	840.784	0.004
11. GLZLM_LGZE	25	0.005	0.027	0.004	0.023	0.026
12. GLZLM_HGZE	25	426.704	582.915	258.133	324.782	0.026
13. GLZLM_SZLGE	25	0.003	0.015	0.002	0.013	0.016
14. GLZLM_SZHGE	25	242.679	348.291	175.596	172.695	0.033
Radiomics that showed a significant difference between EMNT vs. peritumoral edema						
CONVENTIONAL_Kurtosis	28	3.716	4.966	3.454	1.512	0.028
2. CONVENTIONAL_ExcessKurtosis	28	0.716	1.966	0.454	1.512	0.028
3. DISCRETIZED_ Kurtosis	28	3.706	4.956	3.452	1.504	0.034
4. DISCRETIZED_ ExcessKurtosis	28	0.706	1.956	0.452	1.504	0.034
Radiomics that showed a significant difference between tumor tissue versus peritumoral edema						
GLRLM_LRLGE	24	0.007	0.050	0.004	0.046	0.032

IQR: Interquartile range. * Bivariate assessment with the Mann–Whitney U test.

## Data Availability

The data used to support this study’s findings are available from the corresponding author upon reasonable request.

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
