# Peer review of "Identification of Radiomic Signatures in Brain MRI Sequences T1 and T2 That Differentiate Tumor Regions of Midline Gliomas with H3.3K27M Mutation"

_diagnostics, 2023, doi:10.3390/diagnostics13162669_

Round 1
Reviewer 1 Report
Dear Authors,
Thanks!
Institutional emails:
Institutional emails refer to emails that require a confirmed identity, for instance, a university staff member (Researcher) or Professor.
Ins this sense, please, provided your institution email.
Study title:
A good title contains the fewest possible words that adequately describe the contents and/or purpose of your research pape rand summarizes the main idea or ideas of your study.
Please, see: https://www.ncbi.nlm.nih.gov/pmc/articles/PMC6398294/
So, it should not be too long or too short (or cryptic)
Abstract:
Patients and methods.
“Statistical analysis included normal distribution tests and the Mann-Whitney U test”
Please, insert IBM® SPSS® Statistics – version ? significance level?
Results
Table 2 and Table 3.
N = population
n = sample
Please, insert n
Discussion
I would strongly "advise" the authors of this paper to rewrite their discussion to produce a more contextualised discussion toward a clear purpose.
Conclusion:
Please, insert:
· Pratical implications
References
The references must follow the guidelines.
For instance:
13. Federación, D.O.d.l. NORMA Oficial Mexicana NOM-012-SSA3-2012, Que establece los 445 criterios para la ejecución de proyectos para la investigación de salud en seres humanos. 446 Available online: 447 https://dof.gob.mx/nota_detalle.php?codigo=5284148&fecha=04/01/2013#gsc.tab=0 (accessed 448 on November 16th).
14. Assembly, t.W.G. WMA DECLARATION OF HELSINKI – ETHICAL PRINCIPLES FOR 450 MEDICAL RESEARCH INVOLVING HUMAN SUBJECTS. Available online: 451 https://www.wma.net/policies-post/wma-declaration-of-helsinki-ethical-principles-for- 452 medical-research-involving-human-subjects/ (accessed on November 11th ). 453
15. Wen, P.Y.; Huse, J.T. 2016 World Health Organization Classification of Central Nervous 454 System Tumors. Continuum (Minneap Minn) 2017, 23, 1531-1547, 455 doi:10.1212/CON.0000000000000536.
47. LIFEx. LIFEx Analytics. Available online: https://www.lifexsoft.org/index.php/product/lifex- 565 analytics (accessed on june 21st).
Thank you for considering my suggestions, and I look forward to seeing your revised manuscript.
Sincerely,
Referee
-
Reviewer 2 Report
The title indicates adequately the study design.
The abstract provides an informative and balanced summary of the study.
The manuscript is well structured with an adequate and clear presentation of scientific structure, material and methods, results, discussion, conclusion and limitations.
The cited references contains recent publications.
Images and tables are well explained and sufficient.
The language is fluid and clear.
Methods and results are well explained.
The discussion is fluid, and the passages are clear.
The final conclusions are consistent.
Reviewer 3 Report
It is an exciting study for neurooncologists.
H3K27M characteristically exists in diffuse intrinsic pontine glioma (DMG), and its expression can cause a global decrease in histone methylation at the lysine residue. Textural and shape analysis is gaining considerable interest in medical imaging, particularly to identify parameters characterizing tumor heterogeneity and to feed radiomic models. The employed multiplatform freeware LIFEx enables calculating conventional, histogram-based, textural, and shape features from PET, SPECT, MR, CT, US images or any combination of imaging modalities.
The goal of this manuscript is „to identify novel quantitative imaging biomarkers represented by radiomics that can complement the assessment of this pathology and allow a deeper information acquisition than medical images alone. This is particularly helpful in cases where a biopsy cannot be performed, which is common in midline gliomas.”
The authors also aim to identify radiomic signatures valid for discriminating tumour regions in DMG brain MRI sequences post-gadolinium T1 and T2, precisely characterizing patients with the H3.F3K27M mutation.
The results are the following: EMNT vs Tumour: From the T1 sequence, ten radiomics were identified, and 14 radiomics from the T2 sequence, but only one radiomics identified viable tumours in both sequences (p < .05) 49 (DISCRETIZED_Q1). Peritumoral oedema vs EMNT: From the T1 sequence, five radiomics were identified, and four radiomics from the T2 sequence. However, four radiomics could discriminate peritumoral edema in both sequences (p < .05) (CONVENTIONAL_Kurtosis, CONVEN-52 TIONAL_ExcessKurtosis, DISCRETIZED_Kurtosis, and DISCRETIZED_ExcessKurtosis). There were no radiomics useful for distinguishing tumour tissue from peritumoral oedema in both sequences.
The authors were so able to identify radiomics that can discriminate different regions of DMG tumours (EMNT, viable tumour, and peritumoral oedema) in MRI-derived T1 and T2 sequences. The authors claim that obtaining radiomics features that can discriminate between healthy tissue and tumour components could be very useful in multimodal treatment planning in clinical practice. This includes the evaluation of these significant radiomics in patients that cannot be biopsied, supporting the evaluation and decision tree.
However, the ability of radiomics to differentiate DMG from other brain tumours is not evident. If the authors plan Radiomics introduced into clinical practice, they should answer this problem.
The authors should clearly explain to the readers what radiomics offers in everyday clinical practice.
What do the authors mean by … H3.3K27M mutation DICOM files … in the sentence: „Patients and methods. Retrospective study. From a database of 126 DMG patients (children, adolescents, 43 and young adults), only 12 had available brain magnetic resonance DICOM and H3.3K27M mutation DICOM files.”
Two expert neuro-oncologists drew ROIs manually across all slices of the selected sequences individually (T1 post-gadolinium and T2), where the three tumour regions were present. Figure 3 does show an example of the MRI appearance of DMG but not what a typical ROI looks like.

Round 2
Reviewer 1 Report
Dear Authors,
Thanks!
Kind regards
-